# Acute Exposure to Glycated Proteins Impaired in the Endothelium-Dependent Aortic Relaxation: A Matter of Oxidative Stress

**DOI:** 10.3390/ijms232314916

**Published:** 2022-11-29

**Authors:** Sarah D’Haese, Dorien Deluyker, Virginie Bito

**Affiliations:** Uhasselt, Cardio & Organ Systems (COST), Biomedical Research Institute, Agoralaan, 3590 Diepenbeek, Belgium

**Keywords:** acute exposure, high molecular weight advanced glycation end products, aorta, endothelium-dependent vasorelaxation, oxidative stress, superoxide dismutase

## Abstract

Chronically increased levels of high molecular weight advanced glycation end products (HMW-AGEs) are known to induce cardiovascular dysfunction. Whether an acute increase in HMW-AGE levels affects vascular function remains unknown. In this study, we examined whether acute exposure to HMW-AGEs disturbs aortic vasomotor function. Aortae were obtained from healthy male rats and were acutely pre-treated with HMW-AGEs in organ baths. Aortic relaxation responses to cumulative doses of acetylcholine (ACh), in the presence or absence of superoxide dismutase (SOD), were measured after precontraction with phenylephrine (PE). Furthermore, levels of 3-nitrotyrosine were evaluated on aortic paraffine sections. In our study, we show that acute exposure to HMW-AGEs significantly decreases the aortic relaxation response to ACh. SOD pre-treatment prevents acute HMW-AGEs-induced impairment by limiting superoxide formation. In conclusion, our data demonstrate that acute exposure to HMW-AGEs causes adverse vascular remodelling, characterised by disturbed vasomotor function due to increased oxidative stress. These results create opportunities for future research regarding the acute role of HMW-AGEs in cardiovascular dysfunction.

## 1. Introduction

Advanced glycation end products (AGEs) are complex compounds formed by the irreversible glycation of amino acids, peptides, or proteins [1]. They are abundantly present in our Western diet, especially in processed foods with high sugar content, and accumulate inside tissues with aging [2,3]. Growing evidence demonstrates that AGEs play an essential role in vascular dysfunction, which is a major risk factor for the development of cardiovascular diseases (CVD) [4,5]. Due to a great variety in the mechanisms for their formation, different types of AGE molecules have been identified [1]. As such, they can be classified based on their ability to form protein cross-links and/or to emit fluorescence [6]. Moreover, AGEs can be distinguished by their molecular weight [1,7,8]. Low-molecular-weight AGEs (LMW-AGEs, <12 kDa) are considered to exist as free proteins or peptide-bound molecules, such as glycated peptides. The remaining AGEs are defined as high-molecular-weight AGEs (HMW-AGEs, >12 kDa) and appear as protein-bound proteins.

In clinical settings, elevated levels of circulating LMW-AGEs are considered early indicators of cardiovascular disease [9]. Especially on the vascular level, they have been shown to correlate with micro- and macrovascular complications in both diabetic and non-diabetic individuals [10,11]. LMW-AGEs can affect vascular function by two mechanisms [4]. First, they cross-link vascular extracellular matrix (ECM) proteins, thereby directly affecting vascular stiffness [12]. In particular, the cross-linking of the collagen by LMW-AGEs prevents its degradation by matrix metalloproteinases, causing increased collagen deposition and, thus, fibrosis in the vessel wall. Secondly, LMW-AGEs activate their receptor for advanced glycation end products (RAGE) expressed on vascular endothelial cells, initiating several intracellular signalling pathways [11,13]. In particular, the interaction between LMW-AGEs and RAGE activates nicotinamide adenine dinucleotide phosphate (NADPH) oxidase, resulting in the formation of oxidative stress. The excessive generation of reactive oxidant species inside the vascular endothelial cells triggers the activation of nuclear factor kappa beta (NF-κβ), linking LMW-AGEs to the inflammatory pathways [14,15]. In detail, NF-κβ induces the transcription of pro-inflammatory genes interleukin 6, interleukin 1β, and tumour necrosis factor alfa. Via a pathway of cross-talk, the interaction between LMW-AGEs and RAGE also promotes the production of transforming growth factor β, the master regulator of fibrosis, leading to an increased synthesis of vascular ECM proteins [14,16,17].

Although anti-AGE therapies have been found to be effective in preclinical models with increased LMW-AGEs, their clinical values are yet suboptimal, suggesting the presence of other AGE entities [18,19]. Previous research has mainly focused on LMW-AGEs (e.g., carboxymethyllysine (CML)), as they display high stability allowing simple detection methods, while the involvement of HMW-AGEs in CVD remains largely unknown [1]. Interestingly, Miura et al. reported that non-CML AGEs correlate with the severity of microvascular complications in diabetic patients [20]. In addition, HMW-AGEs have been shown to bind more easily with RAGE than LMW-AGEs in diabetic nephropathy, showing their higher pathogenic potential [21]. Recently, Fan et al. demonstrated that HMW-AGEs exert a higher detrimental effect on hepatic glucose metabolism than LMW-AGEs [22]. Altogether, there is growing evidence to show that HMW-AGEs are key players in the development and progression of different diseases, including CVD [7]. In this context, our research group previously showed that a chronic increase in HMW-AGE levels in vivo leads to cardiac dysfunction in rats, characterised by interstitial fibrosis, myocardial wall hypertrophy, and increased left ventricular pressure [23]. At the cellular level, chronic exposure to high levels of HMW-AGEs induces cardiomyocyte remodelling, resulting in impaired cellular function [24]. In addition, we have previously shown that the chronic administration of HMW-AGEs disturbs the vasomotor balance in rat aortae which is the result of increased oxidative stress [25].

Nevertheless, the acute effect of HMW-AGEs, representing their exposure within minutes of up to 24 h, on the cardiovascular system remains partially unclear. Our group has demonstrated that acute exposure to HMW-AGEs reduces cardiomyocyte fractional cell shortening in vitro [26]. However, to date, it remains unknown whether acute exposure to HMW-AGEs influences the vascular function of the aorta. To address this research question, we examined the vasomotor function in isolated rat aortic rings acutely exposed to HMW-AGEs.

## 2. Results

### 2.1. Acute Exposure to HMW-AGEs Alters the Vasomotor Function

To examine endothelium-dependent and –independent relaxation after acute HMW-AGEs exposure, relaxation responses to cumulative doses of acetylcholine (ACh) and sodium nitroprusside (SNP), respectively, were measured in aortic rings ex vivo. ACh-induced relaxation was significantly impaired after 30 min of HMW-AGEs exposure compared to control rings non-exposed to HMW-AGEs (Figure 1A).

In line with these data, exposure to HMW-AGEs significantly lowered E_max_ to ACh compared with the control rings (HMW-AGEs: 71.2 ± 3.0 % in HMW-AGEs vs. 85.1 ± 1.9 % in non-exposed, *p* < 0.05, Table 1). In addition, HMW-AGEs significantly increased the logEC_50_ compared to the control (−6.54 ± 0.07 M in HMW-AGEs vs. −6.81 ± 0.07 M in non-exposed, *p* < 0.05, Table 1). Nevertheless, except at one low dose of SNP, there were no significant differences in the relaxation response to SNP between the groups (Figure 1B), resulting in unchanged E_max_ and logEC_50_ values for SNP between the groups (Table 1).

The vascular contraction after HMW-AGEs was assessed by evaluating the contraction upon phenylephrine (PE) stimulation (Figure 1C). Acute HMW-AGEs exposure did not affect E_max_ obtained with PE (Table 1). However, logEC_50_ for PE in the HMW-AGEs-treated rings was significantly decreased compared to the value of the non-exposed rings, indicating an increased vascular reactivity after HMW-AGEs exposure (*−*7.4 ± 0.1 M in HMW-AGEs vs. *−*7.1 ± 0.1 M in non-exposed, *p* < 0.05, Table 1).

### 2.2. Preincubation with Superoxide Dismutase Improves Endothelium-Dependent Relaxation after Acute HMW-Ages Exposure

To investigate the underlying mechanism resulting in acute HMW-AGEs-impaired relaxation, the aortic rings of all the groups were exposed to antioxidant superoxide dismutase (SOD) for 30 min. Following co-incubation SOD and HMW-AGEs, dose–response curves to ACh were acquired. As expected, we found no differences in the relaxation responses to ACh between the non-exposed group and its SOD preincubated equivalent (Figure 2A). Accordingly, no changes in the E_max_ and logEC_50_ values for ACh between the two groups were found (Table 1). In aortic rings preincubated with HMW-AGEs, SOD improved relaxation responses to ACh at all concentrations of ACh (10^−8^ M ACh: *p* < 0.01, 10^−7^ M ACh: *p* < 0.001, 10^−6^ M ACh: *p* < 0.001, 10^−5^ M ACh: *p* < 0.001, Figure 2B). In line with these findings, SOD preincubation significantly increased E_max_ in aortic rings which were acutely exposed to HMW-AGEs (93.9 ± 1.50% in HMW-AGEs + SOD vs. 71.2 ± 3.0% in HMW-AGEs, *p* < 0.05, Table 1). Furthermore, the logEC_50_ value of the HMW-AGEs group significantly decreased after SOD pre-treatment (−7.4 ± 0.1 M in HMW-AGEs + SOD vs. −6.54 ± 0.1 M in HMW-AGEs, *p* < 0.05, Table 1).

To identify whether increased oxidative stress mediates the relaxation impairment in HMW-AGEs incubated aortae, 3-nitrotyrosine (3-NT) was measured in aortic tissue sections. Our data shows the clear staining of 3-NT in the endothelium (intima) of aortic tissue from acutely HMW-AGEs-incubated aortic rings (Figure 2C). Moreover, the quantification revealed a three-fold increase in aortic 3-NT staining compared to non-exposed (27.3 ± 2.3% in HMW-AGEs vs. 8.8 ± 1.8% in non-exposed rings, *p* < 0.001, Figure 2D). SOD preincubation in HMW-AGEs rings significantly decreased the levels of 3-NT to the control level (27.3 ± 2.3% in HMW-AGEs vs. 6.5 ± 0.7% in HMW-AGEs + SOD, *p* < 0.001, Figure 2D).

## 3. Discussion

Current research mainly focuses on the chronic role of advanced glycation in cardiovascular dysfunction and has demonstrated its deleterious role in disease progression [27]. Nevertheless, Greven et al. recently demonstrated that AGEs immediately accumulate in intensive care unit patients, implying that these acute changes might determine critical clinical outcomes [28]. Thus, investigation of the acute effects of AGEs on cardiovascular function is essential to completely understand the pathophysiological process of AGEs. In this study, we demonstrate that acute exposure to HMW-AGEs impairs endothelium-dependent relaxation by increasing superoxide formation.

### 3.1. Acute Exposure to HMW-AGEs Impairs Endothelium-Dependent Relaxation

The endothelium represents a dynamic compartment of the vessel as it tightly regulates the vascular tone by secreting several paracrine factors, which are able to regulate the contraction and relaxation of the smooth muscle cells [29]. Particularly, nitric oxide (NO) is the main regulator of endothelium-dependent vasorelaxation [30]. Physiologically, NO production is initiated when ACh binds its muscarinic, G_q_-coupled receptor (AChR) on the vascular endothelium [31]. This interaction subsequently up-regulates phospholipase C, which in its turn, transforms phosphatidylinositol 4,5- bisphosphate into inositol triphosphate (IP_3_). When IP_3_ binds its receptor on the SR membrane, calcium (Ca^2+^) is released in the cytosol. The following complex formation between Ca^2+^ and calmodulin (CaM) triggers endothelial nitric oxide synthase (eNOS) to convert its substrate L-arginine into NO. Subsequently, the vasodilator diffuses through the extracellular medium into the vascular smooth muscle cells. There, NO promotes guanylyl cyclase to produce cyclic guanosine monophosphate (cGMP) from guanosine triphosphate. cGMP-mediated stimulation of protein kinase G eventually causes intracellular Ca^2+^ concentration in the smooth muscle cells to decrease, resulting in muscle relaxation. Alternatively, endothelium-independent vasorelaxation can be studied by adding SNP, a NO donor [32]. By the spontaneous release of NO, SNP exerts its relaxant effect directly on smooth muscle cells. 

In this study, we demonstrate that acute exposure to HMW-AGEs impairs endothelium-dependent relaxation in isolated rat aortic rings. Notably, the dose–response curve to ACh undergoes an upward shift in the HMW-AGEs-exposed rings compared to non-exposed control rings. This shift is accompanied by significantly decreased E_max_ and increased logEC_50_. The change in logEC_50_ is an indicator of agonist potency and implies that acute HMW-AGEs accumulation can alter aortic sensitivity for ACh [33]. Accordingly, it is likely that HMW-AGEs can change the protein density of (AChR) on the cellular membrane, thereby affecting its responsiveness [34]. In addition, as AGEs have been demonstrated to cross-link domains of intracellular proteins (e.g., sarco/endoplasmic reticulum Ca^2+^-ATPase), they might influence the tertiary structure of cell membrane receptors as well, thereby affecting their function [35,36]. Indeed, HMW-AGEs-induced cross-linking might decrease the capability of AchR to bind to their ligand, leading to impaired receptor activation and increased logEC_50_ values [33]. Interestingly, HMW-AGEs act via cross-linking, as previously shown by our group in a chronic HMW-AGEs rat model, further strengthening the proposed mechanism [23]. However, this remains hypothetical, and further research is needed. Accordingly, mass spectrometry could be used to confirm whether cross-links are formed on the AChR and identify their location on the protein [36].

In addition, we show that the endothelium-independent relaxation response, as induced by the NO donor SNP, is equal in both groups. Indeed, this unchanged response to SNP does not only confirm the contribution of the endothelium in acute HMW-AGEs circumstances but also indicates that acute increased HMW-AGE levels do not impair the NO-mediated smooth muscle function. Overall, the acute effect of HMW-AGEs is similar to their chronic impact on aortic vasorelaxation, as previously examined by our group [4]. Furthermore, our findings are in line with recent research performed by others regarding the acute effect of AGEs on aortic relaxation. Indeed, Su et al. showed that a 24 h exposure to LMW-AGEs decreases the rat aortic relaxation response to ACh, whereas their relaxation response to SNP remains unchanged [37]. Moreover, Yin et al. have shown that exposure of rat aortic rings to BSA-derived AGEs for 60 min inhibits the endothelium-dependent relaxation associated with a lower E_max_ and higher EC_50_ value [38]. However, which class of AGEs has been applied in their study is unclear. Finally, and also in line with our results on aortic rings exposed to methylglyoxal (MG), a reactive precursor of AGEs for 50 and 60 min also display an impaired endothelium-dependent vasorelaxation [39,40]. The group of Ahmed et al. also confirmed the vascular injury after a 60 min exposure to the AGEs precursor MG [41].

### 3.2. The Impaired Endothelium-Dependent Relaxation Is Caused by Increased Superoxide Formation

Acute HMW-AGEs exposure can impair the aortic vasomotor tone by affecting the bioavailability of NO [30]. Superoxide is able to scavenge NO and, as such, plays a critical role in vascular NO status [42,43]. Interestingly, the accumulation of superoxide has been shown to be the cause of the attenuated endothelium-related relaxation capacity in diabetic rat aortae [44]. Furthermore, our group demonstrated previously that impaired endothelium-dependent relaxation caused by sustained increased levels of HMW-AGEs is mediated by an increased superoxide formation [25]. In the current study, we show that SOD improves the endothelium-dependent relaxation in aortic rings that are acutely preincubated with HMW-AGEs. This finding indicates that the vasorelaxation dysfunction elicited by acute HMW-AGEs exposure is attributable to increased superoxide production. Interestingly, when superoxide radicals react with NO in the vessel, peroxynitrite and its protein-modification 3-NT are excessively formed [45,46]. As shown by the increased expression of 3-NT in the aortic wall of rings acutely preincubated with HMW-AGEs, our current data, indeed, suggest an accumulation of peroxynitrite and, thus, confirm the presumed increased oxidative stress. Altogether, these findings confirm the deleterious role of excess superoxide formation and, according to 3-NT modification in aortic rings after exposure to HMW-AGEs. Our results are in line with the study of Su et al. in which 24 h LMW-AGEs incubation has been shown to elevate aortic superoxide formation, thereby decreasing NO levels and impairing vascular endothelial function [37]. In addition, it has been shown that chronic administration of MG leads to increased levels of 3-NT in the aortic walls of rats and, therefore, impairs vasorelaxation [42]. Regarding the underlying mechanisms, it is known that AGEs stimulate superoxide formation via NADPH oxidase after RAGE activation [47,48,49]. Nevertheless, our group found that HMW-AGEs do not mediate their effects via RAGE interaction but, likely, via cross-linking [23]. Future research should reveal whether acutely elevated HMW-AGE levels stimulate superoxide formation via RAGE-dependent or -independent mechanisms in the vascular endothelium.

However, it should be noted that the vascular NO content also relies on the activity of the NO-producing enzyme eNOS [50]. The recent work of our group showed that the impaired aortic relaxation observed after chronic HMW-AGE exposure is not due to reduced eNOS activity [25]. In that context, the underlying mechanisms of AGEs for vascular dysfunction can differ between acute and chronic circumstances. In this regard, AGEs have been shown to acutely influence eNOS activity by two pathways, namely affecting its expression and decreasing the availability of its cofactors. Concerning the first pathway, two studies have demonstrated that the incubation of vascular endothelial cells with AGEs for 24 h has reduced eNOS mRNA and protein levels [48,51]. Whether acute exposure to HMW-AGEs can affect vascular function by the downregulation of eNOS activity remains to be determined in future experiments. For this, dose–response curves to Ach can be assessed in the presence of an eNOS inhibitor (e.g., nitro-L-arginine methyl ester) in aortic rings, which are acutely exposed to HMW-AGEs ex vivo. To further explore the enzyme’s involvement, the eNOS gene and protein levels can be determined in aortic endothelial cells after acute HMW-AGE exposure. Regarding the second pathway, Naser et al. showed that a 5 min incubation with AGEs causes a depletion of Ca^2+^ from the intracellular store in aortic endothelial cells [52]. Interestingly, Ca^2+^ represents an essential cofactor for adequate eNOS activity [53]. Briefly, inactive eNOS interacts with caveolin-1, a protein that is located in the endothelial caveolae (i.e., small invaginations in the cell membrane) [54]. The inhibitory eNOS-caveolin-1 complex can be dissociated by the binding of Ca^2+^/CaM to eNOS, leading to the activation of the enzyme. Thus, changes in the Ca^2+^ homeostasis of vascular endothelial cells might cause alterations in eNOS-mediated NO production, resulting in vasomotor dysfunction. Accordingly, it might be interesting to acutely expose aortic endothelial cells to HMW-AGEs and measure subsequent intracellular Ca^2+^ alterations using a fluorescent Ca^2+−^ binding dye [55].

### 3.3. Acute HMW-AGEs Exposure Tends to Increase Contractile Vascular Reactivity 

The sympathetic nervous system plays a critical role in the regulation of the aortic vasomotor tone [56]. In particular, the activation of sympathetic nerves by extrinsic stimuli leads to the release of neurotransmitters (e.g., epinephrine), which bind to G_q_-coupled, α_1_-adrenergic receptors on vascular smooth muscle cells [57]. Upon subsequent increase in the intracellular Ca^2+^ concentration, the smooth muscle cells will contract. Alternatively, sympathetic vasoconstrictive responses can also be artificially induced and investigated in preclinical models by the administration of exogenous drugs. PE, a selective agonist for α_1_-adrenergic receptors, is generally known to elicit dose-dependent vasocontraction in isolated rat aortae [58,59]. Our group previously demonstrated that chronic exposure to HMW-AGEs significantly strengthens aortic contraction after cumulative doses of PE [25]. In the current study, we found no significant changes in the PE-induced dose–response curve after acute exposure to HMW-AGEs. This might imply that HMW-AGEs require a longer incubation period to introduce changes in aortic contractions, while endothelium-dependent vasorelaxation can be affected acutely. Indeed, multiple studies showed that rat aortic rings that are exposed to MG for 50- or 60 min and experience exaggerated contractile responses to cumulative doses of PE [39,40,60]. Interestingly, we showed that logEC_50_, the indicator of agonist potency, for PE in the HMW-AGEs-exposed rings is significantly lower than the value of the non-exposed rings. This finding might suggest that acute HMW-AGEs exposure increases the responsiveness of α_1_-adrenergic receptors, possibly by cross-linking the receptor or affecting its membrane protein density.

## 4. Materials and Methods

### 4.1. Animal Experiments

This study was conducted according to the guidelines described in the EU Directive 2010/63/EU for animal experiments and approved by the Local Ethical Committee (Ethical Commission for Animal Experimentation, UHasselt, Diepenbeek, Belgium; ID 202051K). All rats were group housed under temperature- and humidity-controlled conditions (21 °C, 60% humidity) on a 12–12 h light-dark schedule. Animals were fed a standard pellet diet and had access to water ad libitum.

Healthy male Sprague Dawley rats (Charles River Laboratories, L’Arbresle, France) were used. In total, 22 rats were used for the evaluation of aortic vasomotor function, and 12 rats were used for immunohistochemical staining.

### 4.2. Preparation and Protein Concentration Determination of HMW-AGEs

HMW-AGEs were prepared based on the method described by Deluyker et al. [23]. In short, bovine serum albumin (BSA, 7 mg/mL) was incubated with glycolaldehyde dimers (90 mM, Sigma-Aldrich, Diegem, Belgium) in phosphate-buffered saline (PBS, pH 7.4) for five days at 37 °C (i.e., BSA-derived HMW-AGEs). The solution was dialysed against PBS to eliminate the unreacted glycolaldehyde. Sterile syringe filters were used to remove pathogens (0.2 µm sterile filter, Sarstedt, Essen, Belgium). Protein concentrations of the BSA-derived HMW-AGEs (MW > 50 kDa) were assessed with the Pierce BCA Protein Assay kit (Thermo Fisher, Erembodegem, Belgium). The solutions were stored at −80 °C before use.

### 4.3. Assessment of Aortic Vasomotor Function

#### 4.3.1. General Procedure

The rats were sacrificed with an overdose of Dolethal (150 mg/kg IP) and injected with heparin (1000 u/kg IP) to prevent clotting. The thoracic aorta was excised and placed immediately in ice-cold Krebs solution (in mM: 118.3 NaCl, 4.7 KCl, 5.5 glucose, 1.2 MgSO_4_, 1.2 KH_2_PO_4_, 2.5 CaCl_2_, 25 NaHCO_3,_ and 0.026 EDTA; pH 7.45). Adherent perivascular adipose and connective tissue were removed from the vessel, with special care taken to avoid endothelial overstretching. The aortic segment was cut into rings of 3 mm in length. The aortic rings were suspended horizontally between two steel hooks in organ baths filled with continuously heated (37 °C) and oxygenated Krebs solution. One hook was connected to a fixed anchor, whereas the other was extended to a force transducer for isometric tension recordings (MLT 050/A, AD Instruments, Spechbach, Germany) and a data acquisition system (Powerlab 4/25T, AD Instruments). The rings were set at 8 g passive tension to induce optimal stretching, as previously determined, and were allowed to equilibrate for 1 h [4]. During this period, the buffer in the organ bath was refreshed every 20 min. After equilibration, the aortic rings were incubated at random with HMW-AGEs (200 µg/mL) for 30 min or left non-exposed (i.e., incubated only with Krebs solution).

#### 4.3.2. Vasorelaxation and Vasocontraction Responses

The aortic rings were precontracted with PE (10^−7^ M). After reaching a stable contraction plateau phase, the relaxation responses to the cumulative doses of ACh (final bath concentrations: 10^−10^–10^−5^ M) or SNP (final bath concentrations: 10^−10^–10^−6^ M; Sigma-Aldrich, Diegem, Belgium) were assessed to evaluate endothelium-dependent and –independent relaxation, respectively. Additionally, to study the role of superoxide radicals, ACh-generated vasodilation was determined in the presence of SOD (150 kU, Sigma-Aldrich) and was also added 30 min before PE precontraction. Dose–response curves were recorded for 4 min after the addition of the previous concentration of ACh or SNP. Relaxation responses to ACh and SNP were expressed as the percentage of relaxation relative to the PE-induced precontraction. To study the vasoconstrictive responsiveness, the contractile responses to cumulative doses of PE (final bath concentrations: 10^−10^–10^−5^ M) were measured. Dose–response curves were recorded for 4 min after the addition of the previous concentration of PE.

### 4.4. Immunohistochemistry

3-NT staining was performed as described by Haesen et al. [4]. In short, heat-mediated antigen retrieval was performed in deparaffinised 8 µm aortic tissue sections using citrate buffer (pH 6), and sections were washed with PBS and used for 3,3′-diaminobenzidine immunostaining. The sections were incubated with a primary antibody against 3-NT (1:100, mouse monoclonal, Ab7048, Abcam) and were diluted in PBS for 1 h at room temperature. A biotinylated anti-mouse antibody (1:100, E0413, Dako) and streptavidin-HRP (1:400, P0397, Dako) were both applied for 30 min at room temperature. After immunostaining, the nuclei were counterstained using haematoxylin and embedded in a DPX mounting medium. Images were acquired using a Leica MC170 camera connected to a Leica DM2000 LED microscope. The level of staining was assessed in six random fields per section using the colour deconvolution plugin in ImageJ software [61] and was expressed as the % of the total surface area of interest. Two operators blinded for group allocation performed the analysis independently.

### 4.5. Statistical Analysis

Statistical analysis was performed with GraphPad Prism (GraphPad Software, version 5.01, San Diego, CA, USA). The normal distribution of the data was tested via the D’Agostino-Pearson normality test. One-way ANOVA was used to compare the data of the immunostaining. Logarithm dose–response curves were fitted by nonlinear regression and were analysed with LabChart software (v 8.1.13, AD Instruments). Data regarding the dose–response curves were compared via a parametric two-way ANOVA, which was adjusted for repeated measures followed by a Bonferroni post hoc correction. The logarithm of the dose required to acquire a half-maximal response (logEC_50_) and the maximal relaxation response (E_max_) was calculated from the nonlinear regression curves and was compared with the extra sum-of-squares F test. The sample size is indicated as ‘*n*’ (number of aortic rings) or ‘*N*’ (number of rats). Data are expressed as mean ± standard error of the mean (SEM). *p* < 0.05 was considered statistically significant.

## 5. Conclusions

In conclusion, this study demonstrates that acute exposure to HMW-AGEs disturbs the vasomotor function of isolated rat aortae, as illustrated by an impaired endothelium-dependent relaxation. In addition, SOD pre-treatment restored the disturbed vascular relaxation induced by HMW-AGEs. The role of superoxide radicals in this mechanism was confirmed by increased levels of aortic 3-NT, a superoxide radical-derived protein modification, after HMW-AGE treatment. Based on these findings, we can conclude that superoxide radicals are the main drivers in the acute, HMW-AGE-induced endothelial dysfunction in the aorta. Overall, this study gains a better understanding of the acute role of HMW-AGEs in cardiovascular dysfunction. Accordingly, the results of this research further emphasis that HMW-AGEs could become new therapeutic targets in cardiovascular disease.

## Figures and Tables

**Figure 1 ijms-23-14916-f001:**
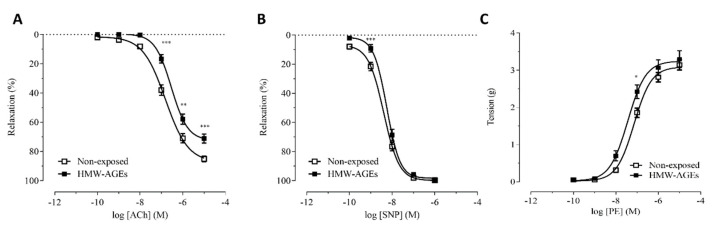
Vasorelaxation and vasocontraction response after acute HMW-AGEs exposure in isolated rat aortic rings. Relaxation response is expressed as the percentage relaxation relative to phenylephrine-induced (final bath concentration: 10^−7^ M) precontraction. (**A**) Relaxation response to cumulative doses of acetylcholine (ACh, final bath concentrations: 10^−10^–10^−5^ M) in aortic rings left non-exposed (*n* = 48, *N* = 15) or preincubated with HMW-AGEs (*n* = 24, *N* = 13). (**B**) Relaxation response to cumulative doses of sodium nitroprusside (SNP, final bath concentrations: 10^−10^−10^−6^ M) in aortic rings left non-exposed (*n* = 47, *N* = 12) or preincubated with HMW-AGEs (*n* = 23, *N* = 12). (**C**) Contractile response to cumulative doses of PE (final bath concentrations: 10^−10^–10^−5^ M) in rat aortic rings which were left non-exposed (*n* = 18, *N* = 4) and preincubated with HMW-AGEs (*n* = 7, *N* = 4). The sample size is indicated as ‘n’ (number of aortic rings) or ‘N’ (number of rats). Data points are visualised as mean ± SEM. * *p* < 0.05, ** *p* < 0.01 and *** *p* < 0.001.

**Figure 2 ijms-23-14916-f002:**
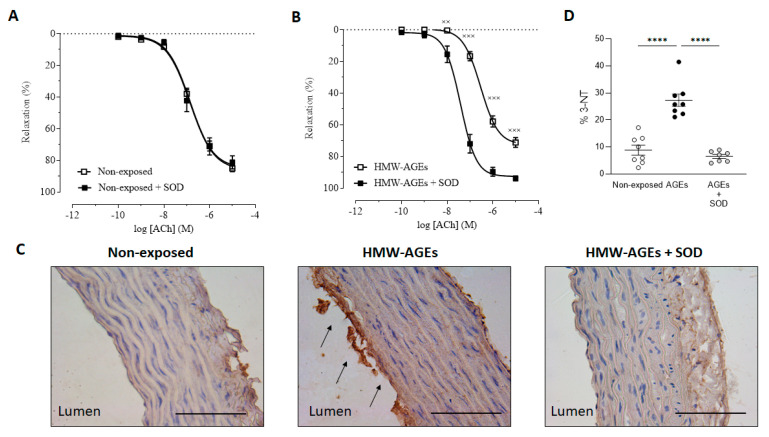
SOD improves endothelium-dependent vasorelaxation, by limiting oxidative stress, in aortic rings after acute HMW-AGEs exposure. Aortic rings were pre-treated with superoxide dismutase (SOD, 150 kU) for 30 min. The relaxation response is expressed as the percentage relaxation relative to phenylephrine-induced (final bath concentration: 10^−7^ M) precontraction. Relaxation response to cumulative doses of acetylcholine (ACh, final bath concentrations: 10^−10^–10^−5^ M) in aortic rings (**A**) Left non-exposed (*n* = 48, *N* = 15) and preincubated with SOD alone for 30 min (*n* = 12, *N* = 3), (**B**) Preincubated with HMW-AGEs (*n* = 24, *N* = 13) and both HMW-AGEs and SOD (*n* = 12, *N* = 7). Data points are visualised as mean ± SEM. ^××^ *p* < 0.01 and ^×××^ *p* < 0.001. (**C**) Representative images of 3-nitrotyrosine (3-NT) staining in transverse aortic tissue sections from non-exposed, HMW-AGEs exposed, and both HMW-AGEs and SOD exposed aortic rings. In aortic rings preincubated with HMW-AGEs, clear staining of the endothelium (intima) was observed (arrows). Scale bars represent 100 µm. (**D**) Quantification of 3-NT from non-exposed (*N* = 8), HMW-AGEs exposed (*N* = 8), and both HMW-AGEs and SOD exposed (*N* = 7) aortic tissue sections. The sample size is indicated as ‘n’ (number of aortic rings) or ‘N’ (number of rats). Data are presented as mean ± SEM. **** *p* < 0.001.

**Table 1 ijms-23-14916-t001:** E_max_ and logEC_50_ values in isolated rat aortic rings.

Groups		E_max_ (%)	LogEC_50_ (M)
ACh	Non-exposed	85.09 ± 1.95	−6.81 ± 0.07
	HMW-AGEs	71.18 ± 2.98 *	−6.54 ± 0.07 *
	Non-exposed + SOD	81.24 ± 4.04	−7.01 ± 0.10
	HMW-AGEs + SOD	93.89 ± 1.50 ^×^	−7.41 ± 0.07 ^×^
SNP	Non-exposed	99.71 ± 0.05	−8.39 ± 0.05
	HMW-AGEs	99.23 ± 0.07	−8.25 ± 0.05
PE	Non-exposed	3.14 ± 0.14	−7.14 ± 0.06
	HMW-AGEs	3.29 ± 0.23	−7.35 ± 0.11 *

The dose required to acquire a half-maximal response (EC_50_) and the maximal relaxation response (E_max_) to acetylcholine (ACh) or sodium nitroprusside (SNP) were assessed from dose–response curves plotted via nonlinear regression. E_max_ is expressed as a percentage of phenylephrine-induced precontraction. LogEC_50_ represents the logarithm of concentration in molar (M). ACh data: aortic rings left non-exposed (*n* = 48, *N* = 15) or preincubated with HMW-AGEs (*n* = 24, *N* = 13), superoxide dismutase (SOD) alone (150 kU; *n* = 12, *N* = 3) or preincubated with both HMW-AGEs and SOD (150 kU; *n* = 12, *N* = 7). SNP data: aortic rings left non-exposed (*n* = 47, *N* = 12) or preincubated with HMW-AGEs (*n* = 23, *N* = 12). PE data: aortic rings were left non-exposed (*n* = 18, *N* = 4) or preincubated with HMW-AGEs (*n* = 7, *N* = 4). The sample size is indicated as ‘n’ (number of aortic rings) or ‘N’ (number of rats). Data are expressed as mean ± SEM. * *p* < 0.05 non-exposed vs. HMW-AGEs. ^×^ *p* < 0.05 HMW-AGEs + SOD vs. HMW-AGEs.

## Data Availability

The data presented in this study are available on request from the corresponding author.

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
