# Peer review of "Acute Exposure to Glycated Proteins Impaired in the Endothelium-Dependent Aortic Relaxation: A Matter of Oxidative Stress"

_ijms, 2022, doi:10.3390/ijms232314916_

Round 1
Reviewer 1 Report
Major:
This manuscript has been performed based on authors’ previous studies, including self citation in the introduction and discussion. In this view, the mechanisms of the role of HMW-AGEs on cardiovascular dysfunction remain unclear which should be more detailed. It is recommended to assess different works which showed robust mechanisms and to verify if HMW-AGEs have similar effects on cardiovascular dysfunction. Several works are showing the role of TGF-beta, IL-10, NADPH oxidase, oxide nitric, endothelial NO-synthase, and collagen deposition in the context of cardiovascular dysfunction. Those studies may be helpful in the introduction and discussion of the data presented by the authors.
Minors:
In the figures, authors should explain the meaning of n and N in the footnote.
Figure 1 did not show the following groups: Non-exposed + SOD and HMW-AGEs + SOD. Please include it.
Reviewer 2 Report
An acute increase in high molecular weight advanced glycation end products (HMW-AGEs) levels may affect the vascular function. The authors predicted this and made their investigations through monitoring the aortic vasomotor function in isolated rat aortic rings being acutely exposed to HMW-AGEs.
Acute exposure to HMW-AGEs was reported to disturbs the vasomotor function of isolated rat aortae, as confirmed through illustration by impaired endothelium-dependent relaxation.
The investigations are methodical. Methods and Materials sections are explained professionally. Experimental techniques applied are appropriate. Discussion is also made as per needed.
The animal model experiments were done taking permission from appropriate authorities-good. Number of rats was enough.
However there are a few major modifications needed before making final decision as follows:
Introduction is poor. An elaborative efforts are needed. More references must be added.
Conclusion is even poorer. Only two line messages on finding can not qualify as conclusion. You need to present sensible amount of information and materials to also support your concluding messages.
Results section-plotting:
It is recommended to also plot the ratio of effects with that of control results. This way readers will find the results easily understandable.
I would recommend the authors to revise and resubmit.
Reviewer 3 Report
Bibliographical citations must be reviewed to follow the standards of the journal, DOI is missing.
There are 34 citations, 14 are more than 10 years old (39%) and 6 more than 20 (18%). I suggest tht can improve older citations to be more recent.
Round 2
Reviewer 1 Report
Although the new version of the manuscript presents a slight improvement, new mechanisms which should bring new advances in this topic were not performed by the authors.
Reviewer 2 Report
The authors have revised following my earlier recommendations. Good luck!